# Role of direct and indirect social and spatial ties in the diffusion of HIV and HCV among people who inject drugs: a cross-sectional community-based network analysis in New Delhi, India

Steven J Clipman[1]*, Shruti H Mehta[2], Aylur K Srikrishnan[3], Katie JC Zook[1], Priya Duggal[2], Shobha Mohapatra[3], Saravanan Shanmugam[3], Paneerselvam Nandagopal[3], Muniratnam S Kumar[3], Elizabeth Ogburn[4], Gregory M Lucas[1], Carl A Latkin[5], Sunil S Solomon[1,2]*

[1]Department of Medicine, Division of Infectious Diseases, Johns Hopkins School of Medicine, Baltimore, United States; [2]Department of Epidemiology, Johns Hopkins Bloomberg School of Public Health, Baltimore, United States; [3]YR Gaitonde Centre for AIDS Research and Education (YRGCARE), Chennai, India; [4]Department of Biostatistics, Johns Hopkins Bloomberg School of Public Health, Baltimore, United States; [5]Department of Health, Behavior, and Society, Johns Hopkins Bloomberg School of Public Health, Baltimore, United States

*For correspondence:
sclipman@jhmi.edu (SJC);
sss@jhmi.edu (SSS)

Competing interest: See
page 13

Reviewing editor: Jennifer
Flegg, The University of
Melbourne, Australia

## Abstract

**Background:** People who inject drugs (PWID) account for some of the most explosive human immunodeficiency virus (HIV) and hepatitis C virus (HCV) epidemics globally. While individual drivers of infection are well understood, less is known about network factors, with minimal data beyond direct ties.

**Methods:** 2512 PWID in New Delhi, India were recruited in 2017–19 using a sociometric network design. Sampling was initiated with 10 indexes who recruited named injection partners (people who they injected with in the prior month). Each recruit then recruited their named injection partners following the same process with cross-network linkages established by biometric data. Participants responded to a survey, including information on injection venues, and provided a blood sample. Factors associated with HIV/HCV infection were identified using logistic regression.

**Results:** The median age was 26; 99% were male. Baseline HIV prevalence was 37.0% and 46.8% were actively infected with HCV (HCV RNA positive). The odds of prevalent HIV and active HCV infection decreased with each additional degree of separation from an infected alter (HIV AOR: 0.87; HCV AOR: 0.90) and increased among those who injected at a specific venue (HIV AOR: 1.50; HCV AOR: 1.69) independent of individual-level factors (p<0.001). In addition, sociometric factors, for example, network distance to an infected alter, were statistically significant predictors even when considering immediate egocentric ties.

**Conclusions:** These data demonstrate an extremely high burden of HIV and HCV infection and a highly interconnected injection and spatial network structure. Incorporating network and spatial data into the design/implementation of interventions may help interrupt transmission while improving efficiency.

**Funding:** National Institute on Drug Abuse and the Johns Hopkins University Center for AIDS Research.

**eLife digest** Understanding the social and spatial relationships that connect people is a key element to stop the spread of infectious diseases. These networks are particularly relevant to combat epidemics among populations that are hard to reach with public health interventions. Network-based approaches, for example, can help to stop HIV or hepatitis C from spreading amongst populations that use injectable drugs. Yet how social and geographic connections such as acquaintances, injection partners, or preferred drug use places impact the risk of infection is still poorly mapped out.

To address this question, Clipman et al. focused on people who inject drugs in New Delhi, India, a population heavily impacted by HIV and hepatitis C. Over 2500 people were recruited, each participant inviting their injection partners to also take part. The volunteers answered survey questions, including where they used drugs, and provided a blood sample to be tested.

The results showed that, even after adjusting for individual risk factors, where people used drugs and with whom affected their risk of becoming infected with HIV and hepatitis C. In terms of social ties, the likelihood of HIV and hepatitis C infection decreased by about 13% for each person separating a given individual from an infected person. However, geographical networks also had a major impact. Injecting at a popular location respectively increased the odds of HIV and hepatitis C infection by 50% and 69%. In fact, even if the participant was not using drugs at these specific places, having an injection partner who did was enough to increase the risk for disease: for each person separating an individual from the location, the likelihood of being infected with HIV and hepatitis C decreased by respectively 14% and 10%.

The results by Clipman et al. highlight how the relationships between physical spaces and social networks contribute to the spread of dangerous diseases amongst people who inject drugs. Ultimately, this knowledge may help to shape better public health interventions that would take into account the importance of geographical locations.

## Introduction

People who inject drugs (PWID) bear a disproportionate burden of human immunodeficiency virus (HIV) and hepatitis C virus (HCV) infection and account for some of the fastest-growing epidemics globally. While there has been substantial progress in combating these epidemics, HIV and HCV prevalence and incidence among PWID remain high, especially in South, Southeast and Central Asia, and Eastern Europe (*DeHovitz et al., 2014*). Individual-level factors for infection are well established, but less is known about network and spatial drivers of HIV and HCV among PWID, especially from low- and middle-income settings.

Network-based interventions for HIV and HCV are increasingly being implemented; however, they are seldom informed by empirical sociometric network data and more often informed by models (*Zelenev et al., 2018*; *Hellard et al., 2015*; *Rolls et al., 2013*; *Hellard et al., 2014*). Existing data derive from small egocentric network studies of 'indexes' or 'egos' and their immediate connections (first degree 'alters') (*Costenbader et al., 2006*; *Latkin et al., 2009*; *Latkin et al., 2011*; *Latkin et al., 2013*; *Latkin et al., 2010*). Few studies have examined the broader sociometric network, which captures the alters of those first degree alters (second degree alters of the index), and so on, providing a more complete representation of the underlying network (*Figure 1*).

Even less is known about the overlap of these egocentric and sociometric networks in space. While spatial heterogeneity of HIV/HCV burden has been previously described in high-income settings (*Des Jarlais et al., 2018*), less is known about whether transmission is driven more by injection partner connections versus spaces/venues people reside and/or inject within. Incorporating spatial data, specifically in the form of injection venues, can further inform whether independent sociometric injection networks overlap spatially to more comprehensively examine the distribution of HIV/HCV and assess the role of space in the diffusion of disease.

This manuscript aims to characterize egocentric, sociometric, and spatial network structures in a community-based sample of 2512 PWID in New Delhi, India and examine the role of individual- and network-level correlates of HIV and HCV infection.

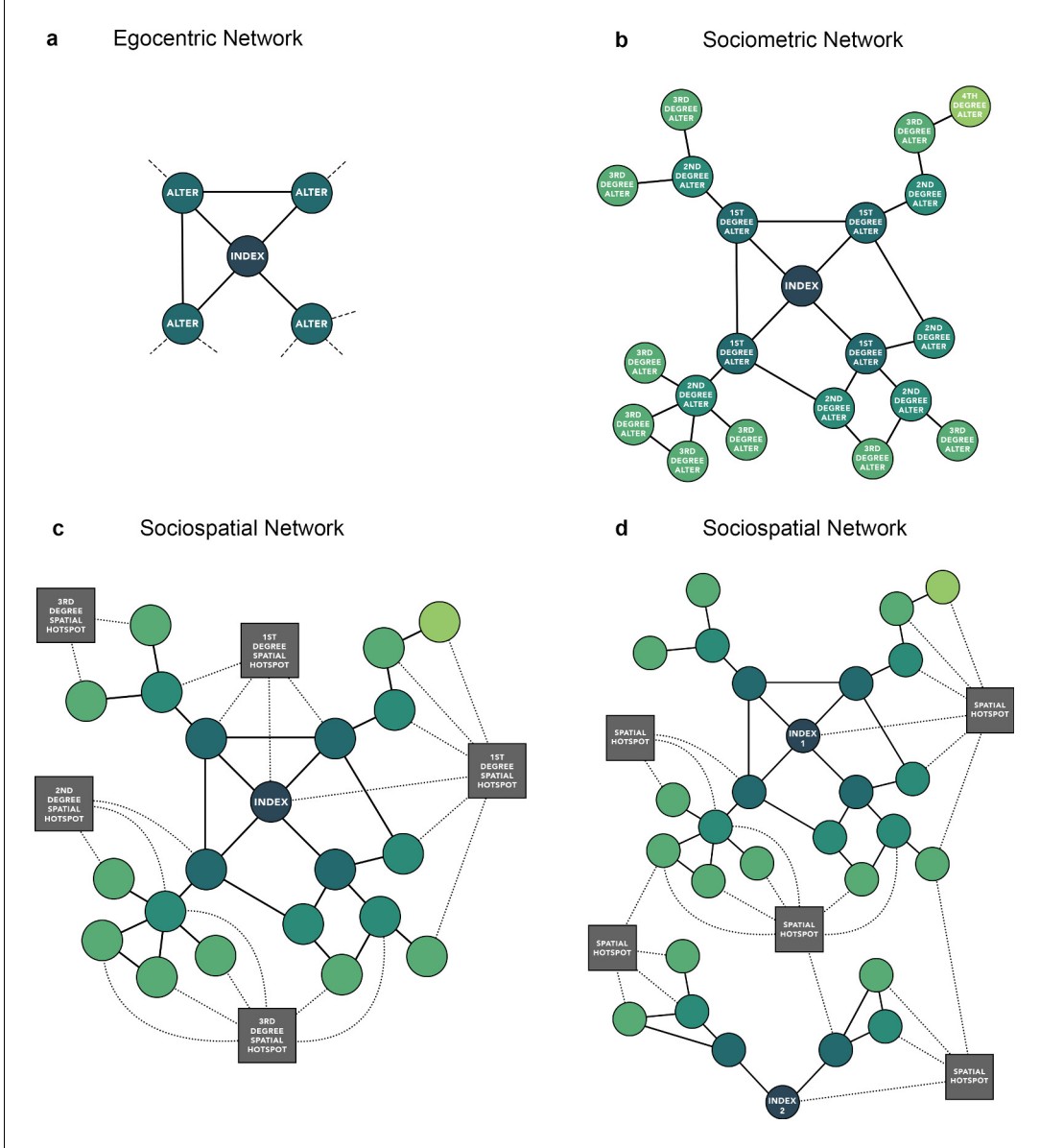

**Figure 1.** Comparison of (**a**) egocentric, (**b**) sociometric, and (**c**) sociospatial network structure. Circular nodes represent an individual. Square nodes represent a venue. Solid edges represent a social tie, dashed lines (**a**) represent a potential tie not captured by an egocentric network, dotted edges represent a spatial tie (**c, d**); (**d**) depicts an example of how two independent sociometric networks can merge into a single network when considering space.

## Materials and methods

### Study overview

The 'Spatial Network Study' is an ongoing dynamic longitudinal cohort of PWID in New Delhi, India established to understand the role of networks in transmission of HIV and HCV among PWID. New Delhi, the capital city of India, is estimated to be home to ~86,000 PWID (*Ambekar et al., 2019*) with HIV prevalence ranging from 13.5% to 35.8% (*Mehta et al., 2015*; *Lucas et al., 2015*) and HCV prevalence ranging from 42.4% to 90% (*Solomon et al., 2015*; *Solomon et al., 2019a*; *Solomon et al., 2019b*). With the exception of index participants, all participants were recruited via a name generator network referral methodology. Participants completed a baseline assessment and were invited to complete semi-annual follow-up visits. This manuscript presents baseline data from this cohort.

## Participant recruitment

Recruitment of the cohort was initiated with two indexes in November 2017—eight more indexes were included later to account for variability in type of drug injected, marital status, and zip code of residence/injection. All 10 indexes were selected from a cross-sectional sample of PWID in New Delhi accrued for an evaluation assessment of a cluster-randomized trial (ClinicalTrials.gov Identifier: NCT01686750) (*Solomon et al., 2019b*). When a participant enrolled in the Spatial Network Study, whether the initial 10 indexes or subsequent recruits, they were asked to recall the names of people with whom they injected in the prior month (regardless of whether they shared injection paraphernalia). In addition, they were asked to provide identifying information about each named network member (e.g., scar on left hand, one finger missing on right hand) and a factoid about each named partner (e.g., 'his wife's name is Priyanka'). Each participant was then provided with a referral card for each named injection partner and was asked to invite them to participate in the study. When these recruits visited the study site, their name and identifying information were compared against the information previously provided. If the information matched, they were enrolled and asked to name, describe, and recruit people with whom they injected in the prior month (recruit's egocentric network and the index's sociometric network). Recruitment continued until the desired sample size (~2500) was reached. Biometric data (fingerprint scans) was used to identify duplicates and establish cross-network linkages (if the same participant was recruited by two different participants). The fingerprint scans were converted to unique hexadecimal codes and stored as described previously (*Solomon et al., 2019b*)—no images were stored.

## Study population

The eligibility criteria varied depending on if the participant was an index or a recruit. Index participants had to (1) be ≥18 years of age, (2) provide written informed consent, (3) report a history of injecting drugs for non-medicinal purposes in the prior 24 months; and (4) have consented to be recontacted from the prior cross-sectional sample in 2016. The eligibility criteria for recruits were (1) ≥18 years of age, (2) provide written informed consent, (3) recruited to participate in the study via a network referral card, (4) match description provided by their recruiter, and (5) not identified as a duplicate participant by biometric (fingerprint) match. Participants under the age of 18 were excluded since the legal age of consent in India is 18 years. There were no exclusions based on gender or sexual identity.

## Study procedures

Baseline study visits began with informed consent and referral card validation that included matching the factoid/identifying characteristic provided by the recruiter, followed by biometric registration and identification of duplicates. Participants that were identified as duplicates, that is, previously enrolled in the cohort, were not enrolled again; however, these data were used to add additional edges (injection partner connections) to the network. Participants then completed the survey and blood draw, followed by rapid HIV and HCV antibody testing on-site with appropriate pre-test counseling and referrals, as applicable. Participants were provided with referral cards to recruit each of their named injection partners into the study. Participants received INR 300 (USD 3.94) as compensation and could earn an incentive of INR 50 (USD 0. 66) per named network partner they referred who was eligible and completed study procedures.

## Data collection

At baseline, participants completed an interviewer-administered electronic survey that captured information on sociodemographics, substance use and risk behavior, sexual risk behaviors and characteristics, social support, quality of life, and access to HIV and HCV services, among others. The survey also captured detailed information about their egocentric injection network and this data was used to generate referral cards. In addition to injection network data, participants were also asked to list venues where they had injected in the prior 6 months. A list of common injection venues (latitude/longitude) was pre-populated and available on maps of Delhi to select from—participants also had the option to add a new venue if they injected at a venue that was not listed.

## Laboratory procedures

On-site rapid HIV antibody testing was performed in line with the current standard of care for HIV diagnosis in India using three different kits: Determine HIV-1/2 (Alere Medical, USA) (Sensitivity: 99.9%; Specificity: 99.8%), First Response HIV card test 1-2-O (Premier Medical, India) (Sensitivity: 100%; Specificity: 99.9%), and Signal HIV-1/2 (Arkray Healthcare, India) (Sensitivity: 100%; Specificity: 100%). Rapid HCV antibody testing was performed using the Aspen HCV One Step Test Device (Aspen Diagnostics, India) (Sensitivity: 99.8%; Specificity: 99.9%). All residual samples were shipped to the central lab in Chennai for RNA quantification and storage. HIV RNA was quantified in all HIV antibody-positive samples using the Abbott HIV-1 RNA Real-Time PCR (Abbott Molecular Inc, Des Plaines, IL, USA) with a lower limit of quantification (LLOQ) of 150 copies/mL. All HCV antibody-positive samples were tested for the presence of HCV RNA with the Real-Time HCV assay (Abbott Molecular Inc, Des Plaines, IL, USA) with an LLOQ of 30 IU/mL.

## Statistical and computational methods

Statistical analyses were carried out in Python (v3.7.3) and R (v3.5.1). Individual and network variables were analyzed for an association with prevalent HIV and active HCV infection (HCV RNA positive) using univariable and multivariable logistic regression. The Boruta (*Kursa and Rudnicki, 2010*) random forest feature selection algorithm was used to explore candidate factors. Variables were considered for inclusion in multivariable models if they held biological/epidemiological significance or had significant associations in univariable models or significant variable importance scores from random forest (p<0.05).

Networks were constructed with Python using NetworkX (*Hagberg et al., 2008*) and network variables, that is, number of infected injection partners (first degree alters) and network distance from an infected alter/venue, were calculated from the sociometric network (containing only person nodes). Network distance from an infected alter was calculated such that a distance of zero signifies a direct connection, a distance of one signifies one uninfected person along the shortest path between a given participant and infected alter. Similarly, for network distance from an injection venue, a distance of zero signifies a direct connection to the venue and a distance greater than zero signifies the number of person nodes along the shortest path between a participant and venue.

Networks were visualized using Gephi (https://gephi.org) and interactive networks were created using Sigma.js (http://sigmajs.org). Spatial nodes in the network were placed by GPS coordinates to be spatially congruent with their geographic position under a Mercator map projection. Person-nodes were placed using a degree-dependent force-directed algorithm.

## Ethical clearance

The study protocol was approved by Institutional Review Boards at Johns Hopkins Medicine (IRB00110421) and the YR Gaitonde Centre for AIDS Research and Education in India (YRG292). All participants provided written informed consent.

# Results

## Network characteristics

2502 PWID were recruited by 10 indexes (total n=2512). A median one referral card was provided to each participant (range: 0–6) and 75% (2437/3244) of referral cards were returned. As recruitment continued, the sociometric networks of 6 out of the 10 index participants merged into one larger network resulting in a total of 5 discrete sociometric networks (*Figure 2*; *Figure 2—figure supplement 1*). The median degree/egocentric network size was 2 (interquartile range [IQR]: 1–3). The sociometric network diameter was 39, and the average path length was 14. Participants identified a total of 181 unique injection venues, defined as any public space where two or more PWID report injecting drugs together in the prior 6 months, spanning a 20-km radius in New Delhi, India. The median number of venues participants reported injecting in the prior 6 months was 3 (IQR: 2–6). The five discrete sociometric networks depicted in *Figure 2* merged into one sociospatial network (*Figure 3*) when accounting for social and spatial ties between participants (interactive version of figure available at https://github.com/sclipman/sociospatial-baseline, copy archived at swh:1:rev: f22127448e931699530d02475043b2279279d67f; *Clipman, 2021*). The sociospatial network

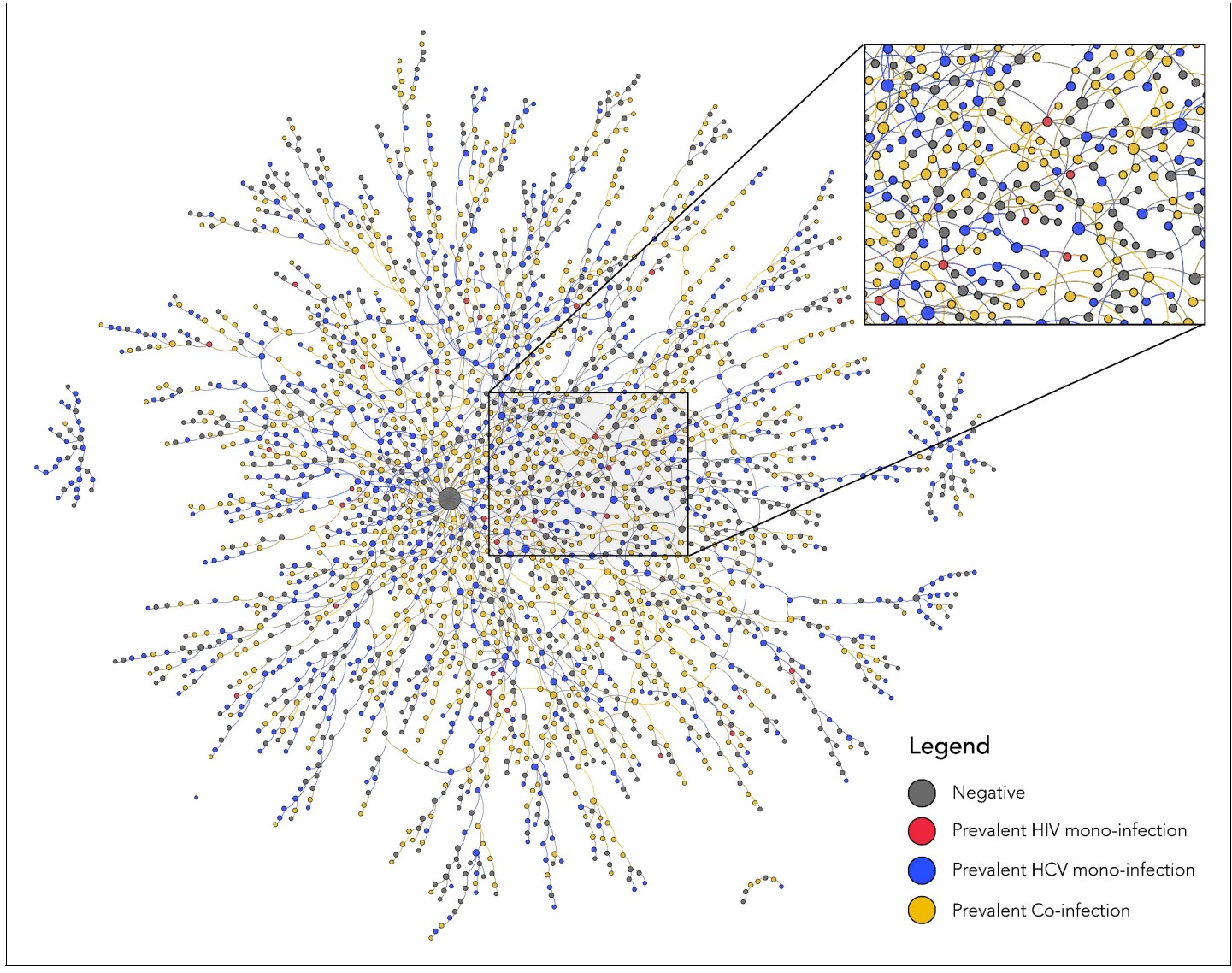

**Figure 2.** Baseline sociometric network structure and HIV/HCV infection status of 2512 people who inject drugs in New Delhi, India. Nodes are colored by infection status and sized by degree. HCV, hepatitis C virus; HIV, human immunodeficiency virus.

The online version of this article includes the following figure supplement(s) for figure 2:

**Figure supplement 1.** Baseline sociometric network structure of 2512 people who inject drugs in New Delhi, India with the 10 indexes that initiated recruitment colored green.

diameter was 8, and the average path length was 3.3, signifying a higher efficiency of network transmission when considering spaces.

## Demographics

The median age of the 2512 participants was 26 years and 2489 (99%) were male—20 cisgender women, and 3 transgender women were recruited (*Table 1*). A total of 218 participants (9%) had at least high school education and 19% reported same-sex behavior. Buprenorphine and heroin were the most commonly injected drugs—2411 (96%) and 1150 (46%) reported ever injecting buprenorphine and heroin, respectively, and 1518 (60%) reported ever sharing injection paraphernalia. The median duration of drug use in the sample was 5 years (IQR: 2–10). 2499 participants reported injecting at least once in the prior 6 months with a median injection frequency of 360 times in the prior 6 months (IQR: 180–540). The demographic and risk characteristics of the indexes are provided as a table in *Supplementary file 1*.

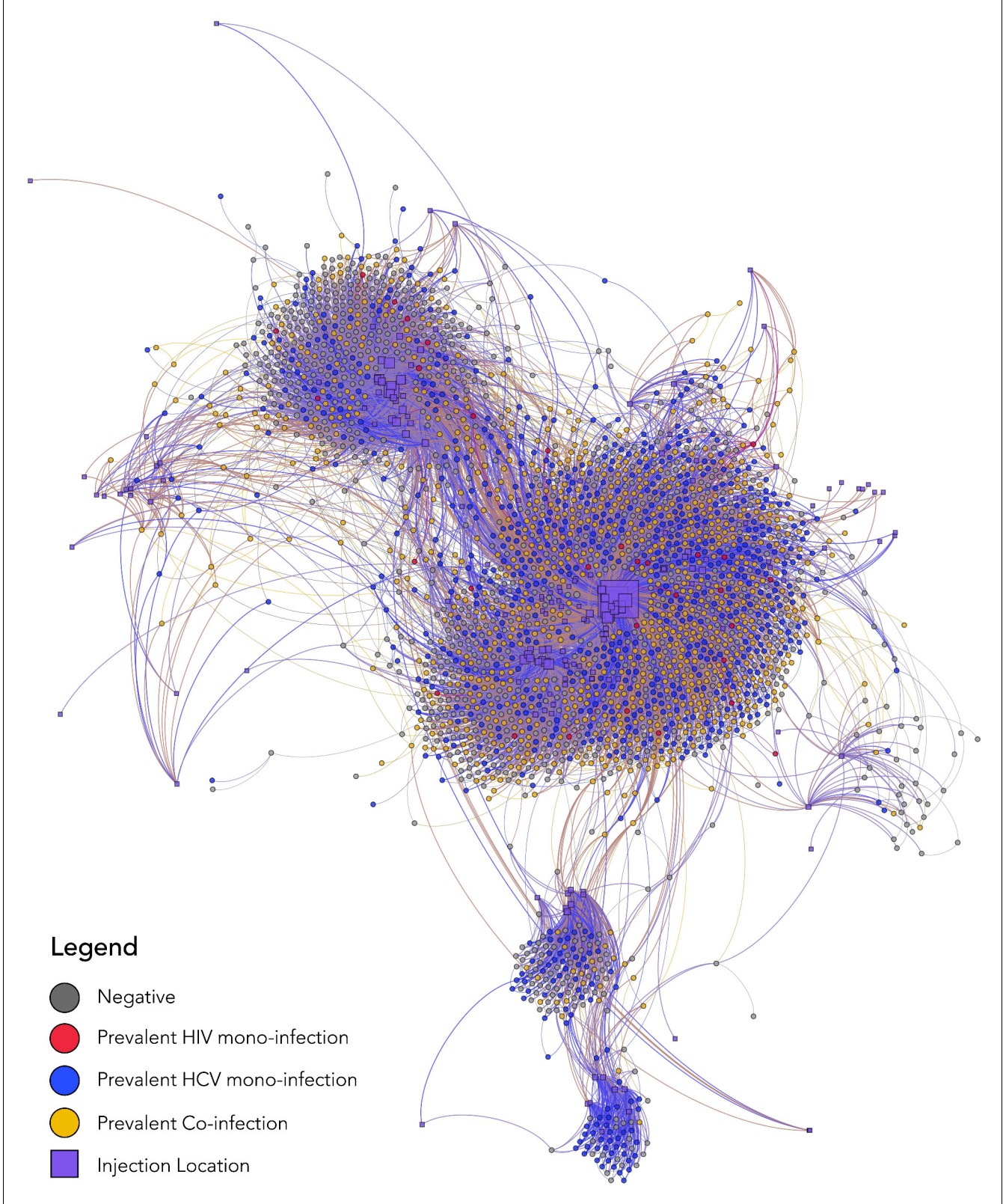

Legend

- ⬤ Negative
- 🔴 Prevalent HIV mono-infection
- 🔵 Prevalent HCV mono-infection
- 🟡 Prevalent Co-infection
- 🟪 Injection Location

**Figure 3.** Baseline sociospatial network structure and HIV/HCV infection status of 2512 people who inject drugs in New Delhi, India. The nodes in this sociospatial network represent persons (circles) or injection venues (squares), and the edges represent a social tie (in the case of a connection between two person nodes) or a spatial tie (in the case of a connection between a person node and spatial node). Person nodes are colored by HIV/HCV

*Figure 3 continued on next page*

*Figure 3 continued*

infection status, sized by degree, and are placed by using a degree-dependent force-directed algorithm. Spatial nodes are sized by degree and placed by GPS coordinates to be geographically congruent under a Mercator projection. HCV, hepatitis C virus; HIV, human immunodeficiency virus.

The online version of this article includes the following figure supplement(s) for figure 3:

**Figure supplement 1.** Geographic extent of 181 injection venues identified by study participants and kernel density plot of (**a**) HIV prevalence and (**b**) HCV prevalence.

**Figure supplement 2.** Distribution of the number of people who report injecting drugs across 181 injection venues in New Delhi, India.

**Table 1.** Participant baseline characteristics of 2512 people who inject drugs in New Delhi, India (parenthesis denote n unless otherwise specified).

|  | Overall | HIV mono-infected | Anti-HCV mono-infected | HIV/anti-HCV co-infected | Negative for HIV and HCV |
|---|---|---|---|---|---|
| Number of participants | 2512 | 31 | 737 | 897 | 847 |
| Median age (IQR) | 26 (22–34) | 24 (20–32) | 27 (22–35) | 26 (22–32) | 26 (22–36) |
| Male gender | 99% (2489) | 97% (30) | 99% (728) | 99% (890) | 99% (837) |
| Self-report of ever having sex with a man | 20% (498) | 26% (8) | 16% (121) | 21% (192) | 21% (177) |
| Self-identify as gay or bisexual | 19% (479) | 23% (7) | 17% (123) | 21% (188) | 20% (167) |
| Highest level of education |  |  |  |  |  |
| *No schooling* | 30% (754) | 32% (10) | 31% (225) | 39% (345) | 21% (174) |
| *Primary school (Grades 1–5)* | 25% (618) | 26% (8) | 25% (183) | 28% (250) | 21% (177) |
| *Secondary school (Grades 6–10) or above* | 45% (1130) | 42% (13) | 44% (327) | 33% (300) | 58% (490) |
| Employment |  |  |  |  |  |
| *Earn daily wage* | 62% (1545) | 68% (21) | 60% (442) | 70% (630) | 53% (452) |
| *Earn weekly or monthly wage* | 28% (714) | 26% (8) | 30% (222) | 21% (190) | 35% (294) |
| *Unemployed* | 7% (165) | 6% (2) | 6% (45) | 5% (47) | 8% (71) |
| Currently Experiencing homeless | 30% (754) | 32% (10) | 30% (221) | 41% (371) | 18% (152) |
| Median years injecting drugs (IQR) | 5 (2–10) | 4 (1–8) | 5 (2–10) | 5 (2–9) | 3 (2–8) |
| Median injections in prior 6 months (IQR) | 360 (180–540) | 360 (360–540) | 360 (180–540) | 360 (344–540) | 340 (96–360) |
| Mean no. injection partners in prior month | 3.2 | 2.9 | 3.3 | 3.5 | 2.8 |
| Ever shared syringes | 60% (1518) | 74% (23) | 63% (463) | 73% (651) | 45% (381) |
| Shared syringes in prior 6 months | 51% (1284) | 65% (20) | 53% (391) | 61% (547) | 39% (326) |
| Type of drug injected (ever) |  |  |  |  |  |
| *Heroin only* | 4% (89) | 6% (2) | 3% (25) | 2% (17) | 5% (45) |
| *Buprenorphine only* | 54% (1350) | 58% (18) | 50% (368) | 47% (422) | 64% (542) |
| *Heroin and buprenorphine* | 42% (1061) | 36% (11) | 46% (342) | 51% (458) | 30% (250) |
| Type of drug injected (prior 6 months) |  |  |  |  |  |
| *Heroin only* | 4% (107) | 7% (2) | 4% (30) | 3% (25) | 6% (50) |
| *Buprenorphine only* | 73% (1820) | 74% (23) | 71% (521) | 70% (630) | 78% (646) |
| *Heroin and buprenorphine* | 22% (559) | 19% (6) | 25% (181) | 27% (240) | 16% (132) |
| Access to services |  |  |  |  |  |
| *Ever tested for HIV* | 48% (1203) | 29% (9) | 53% (394) | 49% (440) | 43% (360) |
| *Ever tested for HCV* | 4% (104) | 0% (0) | 6% (46) | 5% (43) | 2% (15) |
| *Ever used medication assisted therapy* | 36% (906) | 36% (11) | 37% (272) | 32% (290) | 39% (333) |
| *Ever used syringe service program* | 17% (427) | 7% (2) | 19% (137) | 21% (191) | 12% (97) |

*Note*: anti-HCV, HCV antibody.

## Prevalence and correlates of HIV

Baseline HIV prevalence was 37.0% (928/2506), and 92% of these participants had detectable HIV RNA. Out of 928 HIV-positive participants at baseline, 65% were directly connected with at least one other HIV antibody-positive PWID (*Figure 2*); median network distance to another HIV antibody-positive PWID was 0 (range: 0–3). At least one HIV-positive person reported injecting at 155 (86%) of the 181 injection venues identified by participants (*Figure 3—figure supplement 1a*); all participants were directly connected to at least one venue containing an HIV-positive person. Venue #40 was the most frequented injection venue (see *Figure 3—figure supplement 2* for distribution)—1219 (49%) of all participants and 565 (60.0%) of HIV-positive participants reported injecting at this venue. Participants who injected at venue #40 also reported, on average, 32% more injections in the prior 6 months than those who did not report injecting at this venue (p<0.001).

Individual-level variables positively associated with prevalent HIV in multivariable logistic regression included younger age, lower education, experiencing homelessness, decreased sexual activity, sharing syringes, increased injection frequency, and injecting heroin and buprenorphine (*Table 2*). Network-level factors remained highly statistically significant even after adjusting for individual-level correlates. At the egocentric level, odds of prevalent HIV increased by 20% for each additional HIV-positive alter (adjusted odds ratio [AOR]: 1.20; 95% confidence interval [CI]: 1.08–1.34). At the sociometric level, likelihood of HIV infection decreased by 13% with each additional uninfected person between a participant and infected alter (AOR: 0.87; 95% CI: 0.82–0.95). Injecting at venue #40, which represents a participant's immediate spatial network, was positively associated with prevalent HIV (AOR: 1.50; 95% CI: 1.24–1.82) after adjusting for individual- and network-level correlates. The sociospatial network parameter was also independently associated with HIV infection; odds of infection reduced by 14% for each additional person separating a participant from venue #40 (AOR: 0.86; 95% CI: 0.82–0.91). Sociometric and sociospatial network parameters were significantly associated with HIV even after accounting for the egocentric parameter.

## Prevalence and correlates of HCV

Baseline anti-HCV antibody prevalence was 65.1% (1634/2512), and out of 1477 samples with HCV RNA data, 80% had active HCV infection (detectable HCV RNA). The majority of participants were unaware of their HCV status, only 4% (104) reported ever being previously tested for HCV, and nine individuals reported ever testing positive (all received or are currently taking treatment). Therefore, instances where a person had anti-HCV antibodies but no HCV RNA most likely represent natural clearance of HCV infection. A total of 897 (35.7%) participants had evidence of HIV/HCV co-infection (HIV and anti-HCV positive); of these, 658 (73.4%) were HCV RNA positive.

Out of 1634 anti-HCV positive participants at baseline, 86% were directly connected with at least one other anti-HCV positive PWID, and out of 1178 HCV RNA positive participants at baseline, 74% were directly connected with at least one other HCV RNA positive participant. The mean network distance from a participant with active HCV infection (HCV RNA positive) differed significantly by HCV infection status (one-way ANOVA; p<0.001). Among persons with active HCV infection, the mean network distance to another participant with active HCV infection was 0.59 compared to 0.72 for anti-HCV positive persons with undetectable HCV RNA and 0.90 for anti-HCV negative participants. Betweenness centrality was 1.42 times higher on average among the 1178 HCV RNA positive participants compared to the 878 anti-HCV negatives (two-sample t-test; p<0.01). In addition, persons with active HCV infection had significantly higher degree of centrality (p<0.01). A total of 172 (95%) injection venues contained at least one anti-HCV positive person (*Figure 3—figure supplement 1b*); all participants were directly connected to at least one venue containing an anti-HCV positive person. Out of 1219 participants who injected at venue #40, 942 (77.3%) were anti-HCV positive; HCV RNA testing was available on 868, 79.7% of whom had detectable HCV RNA.

Similar individual- and network-level correlates associated with prevalent HIV were associated with active HCV infection (*Table 3*). The odds of HCV RNA positivity increased by 21% with each additional HCV RNA positive first degree alters (AOR: 1.21; 95% CI: 1.10–1.34) and decreased by 10% with each additional uninfected person along the shortest path to an HCV RNA positive participant (AOR: 0.90; 95% CI: 0.82–0.99). Injecting at venue #40 was the strongest correlate, increasing the odds of HCV RNA positivity by 69% (AOR: 1.69; 95% CI: 1.40–2.03), and each additional person between a participant and venue #40 reduced the odds of current HCV infection by 10% (AOR:

**Table 2.** Factors associated with prevalent HIV infection in a sample of 2512 PWID in New Delhi, India.
Columns represent a logistic regression model and depict the odds ratios/adjusted odds ratios and 95% confidence intervals for the included variables.

| Factors associated with prevalent HIV | Univariable OR (95% CI) | Multivariable AOR (95% CI) | Multivariable AOR (95% CI) | Multivariable AOR (95% CI) | Multivariable AOR (95% CI) | Multivariable AOR (95% CI) |
|---|---|---|---|---|---|---|
| Age *per 5 year increase* | 0.88 (0.84–0.92) | 0.88 (0.84–0.92) | 0.90 (0.84–0.93) | 0.88 (0.84–0.92) | 0.88 (0.84–0.93) | 0.88 (0.84–0.93) |
| Education | | | | | | |
| *No schooling (ref.)* | 1.00 | 1.00 | 1.00 | 1.00 | 1.00 | 1.00 |
| *Primary school* | 0.80 (0.65–0.99) | 0.89 (0.71–1.13) | 0.92 (0.73–1.15) | 0.93 (0.74–1.17) | 0.95 (0.75–1.20) | 0.95 (0.75–1.20) |
| *Secondary school or above* | 0.43 (0.35–0.52) | 0.54 (0.44–0.67) | 0.55 (0.44–0.68) | 0.56 (0.45–0.69) | 0.57 (0.46–0.70) | 0.57 (0.46–0.71) |
| Experiencing Homelessness | 2.27 (1.91–2.71) | 1.54 (1.27–1.87) | 1.52 (1.25–1.85) | 1.48 (1.22–1.80) | 1.30 (1.07–1.60) | 1.32 (1.07–1.62) |
| Sexual activity *vaginal or anal sex in prior 6 months* | 0.42 (0.35–0.51) | 0.53 (0.44–0.65) | 0.53 (0.43–0.68) | 0.53 (0.43–0.64) | 0.53 (0.43–0.64) | 0.52 (0.43–0.63) |
| Ever shared syringes | 2.34 (1.96–2.78) | 1.78 (1.48–2.15) | 1.76 (1.45–2.12) | 1.76 (1.46–2.13) | 1.73 (1.43–2.10) | 1.75 (1.44–2.11) |
| Injection frequency *per 50 injections in prior 6 months* | 1.10 (1.08–1.11) | 1.06 (1.04–1.08) | 1.06 (1.04–1.08) | 1.06 (1.04–1.08) | 1.06 (1.04–1.08) | 1.06 (1.04–1.07) |
| Type of drug injected (ever) | | | | | | |
| *Buprenorphine only (ref.)* | 1.00 | 1.00 | 1.00 | 1.00 | 1.00 | 1.00 |
| *Heroin only* | 0.56 (0.33–0.94) | 0.58 (0.34–1.01) | 0.59 (0.34–1.03) | 0.58 (0.33–1.01) | 0.61 (0.35–1.07) | 0.61 (0.35–1.06) |
| *Heroin and buprenorphine* | 1.64 (1.39–1.94) | 1.39 (1.16–1.68) | 1.35 (1.12–1.63) | 1.34 (1.11–1.62) | 1.33 (1.10–1.60) | 1.31 (1.08–1.58) |
| Number infected injection partners *per one person increase in anti-HIV-positive injection partners* | 1.25 (1.13–1.37) | – | 1.20 (1.08–1.34) | 1.15 (1.03–1.28) | 1.16 (1.04–1.29) | 1.14 (1.02–1.27) |
| Network distance from an HIV-infected participant | 0.83 (0.78–0.88) | – | – | 0.87 (0.82–0.95) | 0.90 (0.82–0.96) | 0.92 (0.85–0.99) |
| Injecting at venue #40 | 2.22 (1.88–2.62) | – | – | – | 1.50 (1.24–1.82) | 1.10 (0.85–1.43) |
| Network distance from venue #40 | 0.79 (0.75–0.83) | – | – | – | – | 0.86 (0.82–0.91) |

0.90; 95% CI: 0.85–0.97). This sociospatial parameter was statistically significant even after accounting for the egocentric parameter.

## Discussion

In this sample of PWID in New Delhi, India, we observed an extremely high burden of HIV and HCV infection and strong associations between HIV and HCV within not only an individual's immediate egocentric network but also broader sociometric, spatial, and sociospatial networks that incorporate indirect ties. These data are among the first to elucidate sociometric and spatial injection networks of PWID from a low- and middle-income country and provide critical insights into the design of HIV and HCV programming.

Empirical network data among PWID have often been limited to egocentric network data, which only capture information on individuals and direct contacts (*Costenbader et al., 2006*; *Latkin et al., 2009*; *Latkin et al., 2011*; *Latkin et al., 2013*; *Latkin et al., 2010*). These studies have shown that network instability or turnover in PWID's injection partners promote HIV transmission. Limited sociometric data that exist come primarily from non-PWID populations in developed country settings and support the importance of understanding network connections beyond direct ties. For example, a seminal study assessing HIV and STI transmission in Colorado Springs found that HIV risk appeared

**Table 3.** Factors associated with active HCV infection (HCV RNA positive) in a sample of 2512 PWID in New Delhi, India. Columns represent a logistic regression model and depict the odds ratios/adjusted odds ratios and 95% confidence intervals for the included variables.

| Factors associated with active HCV infection (HCV RNA positive) | Univariable OR (95% CI) | Multivariable AOR (95% CI) | Multivariable AOR (95% CI) | Multivariable AOR (95% CI) | Multivariable AOR (95% CI) | Multivariable AOR (95% CI) |
|---|---|---|---|---|---|---|
| Age *per 5 year increase* | 0.96 (0.92–0.99) | 0.97 (0.93–1.01) | 0.97 (0.93–1.01) | 0.97 (0.93–1.01) | 0.97 (0.93–1.01) | 0.97 (0.93–1.02) |
| Education | | | | | | |
| *No schooling (ref.)* | 1.00 | 1.00 | 1.00 | 1.00 | 1.00 | 1.00 |
| *Primary school* | 0.91 (0.73–1.14) | 1.04 (0.82–1.32) | 1.06 (0.84–1.34) | 1.06 (0.84–1.35) | 1.09 (0.86–1.38) | 1.09 (0.86–1.39) |
| *Secondary school or above* | 0.56 (0.46–0.67) | 0.70 (0.57–0.85) | 0.70 (0.57–0.86) | 0.70 (0.57–0.86) | 0.71 (0.58–0.88) | 0.72 (0.59–0.89) |
| Experiencing Homelessness | 1.97 (1.64–2.36) | 1.45 (1.19–1.77) | 1.40 (1.15–1.71) | 1.41 (1.15–1.72) | 1.17 (0.95–1.45) | 1.18 (0.95–1.45) |
| Sexual activity *vaginal or anal sex in prior 6 months* | 0.58 (0.49–0.69) | 0.69 (0.58–0.83) | 0.69 (0.58–0.83) | 0.70 (0.58–0.84) | 0.70 (0.58–0.84) | 0.69 (0.57–0.83) |
| Ever shared syringes | 2.08 (1.75–2.45) | 1.65 (1.38–1.98) | 1.65 (1.38–1.98) | 1.67 (1.40–2.00) | 1.62 (1.35–1.94) | 1.62 (1.36–1.95) |
| Injection frequency *per 50 injections in prior 6 months* | 1.08 (1.06–1.10) | 1.05 (1.03–1.07) | 1.05 (1.03–1.07) | 1.05 (1.03–1.07) | 1.04 (1.02–1.06) | 1.04 (1.02–1.06) |
| Type of drug injected (ever) | | | | | | |
| *Buprenorphine only* | 1.00 | 1.00 | 1.00 | 1.00 | 1.00 | 1.00 |
| *Heroin only* | 0.80 (0.51–1.24) | 0.85 (0.53–1.35) | 0.85 (0.53–1.35) | 0.84 (0.53–1.34) | 0.91 (0.57–1.45) | 0.90 (0.56–1.44) |
| *Heroin and buprenorphine* | 1.75 (1.48–2.07) | 1.48 (1.23–1.78) | 1.45 (1.20–1.74) | 1.45 (1.21–1.75) | 1.43 (1.18–1.72) | 1.41 (1.17–1.70) |
| Number infected injection partners *per one person increase in HCV RNA positive injection partners* | 1.25 (1.14–1.36) | – | 1.21 (1.10–1.34) | 1.13 (1.00–1.28) | 1.12 (0.99–1.27) | 1.11 (0.98–1.26) |
| Network distance from an HCV RNA+ participant | 0.82 (0.77–0.80) | – | – | 0.90 (0.82–0.99) | 0.93 (0.84–1.03) | 0.96 (0.85–1.00) |
| Injecting at location #40 | 2.29 (1.94–2.70) | – | – | – | 1.69 (1.40–2.03) | 1.31 (1.03–1.68) |
| Network distance from location #40 | 0.81 (0.77–0.84) | – | – | – | – | 0.90 (0.85–0.97) |

low based on individual-level or egocentric network data, but sociometric data revealed risk to be higher than anticipated, with most individuals being within a few steps of an HIV infected person (*Klovdahl et al., 1994*; *Rothenberg et al., 1995*). Other studies in the United States have shown that sociometric networks can propagate HIV, with core individuals of large network components serving as centers of high-risk behavior and pockets of infection that could be targeted by network-based interventions (*Friedman et al., 1997*; *Young et al., 2013*).

The sociometric network presented here provides further support for transmission within large network components in a low- and middle-income country setting. For both prevalent HIV and active HCV infection, we found that while having direct ties with HIV/HCV infection was associated with prevalent HIV/HCV, sociometric factors such as network distance to an infected alter were also independently associated even after accounting for direct ties. Among HCV RNA positive persons, the average sociometric network distance to another HCV RNA positive participant was significantly shorter compared to those of anti-HCV positive persons with undetectable HCV RNA or anti-HCV negative persons, supporting that network proximity to PWID with HCV RNA infection indicates higher likelihood of reinfection.

These analyses further contribute to available network data by overlaying sociometric network data with information on injection venues. Prior studies, including some among PWID, have shown that physical spaces play an important role in the spread of disease, but have not examined spread

through sociometric injection networks or associations with indirect connections to spaces (*Zelenev et al., 2016*; *Gesink et al., 2014*; *Rudolph et al., 2017*; *Logan et al., 2016*). The incorporation of space accounts for undocumented connections between participants in a space as well as spatial factors themselves (e.g., access to harm reduction services). Logan et al. demonstrated this in a sample of 600 participants (including 303 PWID) from Winnipeg, Canada, defining a geographically and socially cohesive community through which infections spread and identifying key venues driving such spread (*Logan et al., 2016*). This is one of the few studies that integrated space with injection network data but represents a small sample from a developed country setting where HIV burden was significantly lower and the social context of drug use is very different.

The idea that the sociospatial risk network may be the most relevant for transmission of disease is particularly salient with the increased focus on network-based interventions to reduce HIV and HCV transmission and optimize care outcomes while conserving resources. These findings suggest that in some settings, the sociospatial network may explain the majority of disease spread and interventions targeted at key spaces have the potential to interrupt transmission across a network and impact an entire city. For example, in this sample, it could be hypothesized that blanket coverage of venue #40 with treatment and pre-risk exposure prophylaxis, could impact transmission across New Delhi given the strong association of venue #40 with prevalent HIV among PWID in New Delhi.

Traditionally network-based interventions have relied heavily on social diffusion. For example, for HIV, network members are used to improve retention to antiretroviral therapy and improve viral suppression (*Klovdahl et al., 1994*), and 'deep chain' respondent-driven sampling is being used to identify undiagnosed or out-of-care HIV-positive men who have sex with men in the United States (*Rothenberg et al., 1995*). For HCV, egocentric network-based treatment approaches have been identified as the optimal approach to deliver therapy while minimizing reinfection (*Rolls et al., 2013*; *Hellard et al., 2014*). However, if networks are highly interconnected especially within other PWID at a venue as observed in New Delhi, failure to incorporate space in the consideration of a network could result in high rates of HCV reinfection.

In addition to network-level factors, these findings further reinforce the importance of well-established individual-level factors such as needle sharing, injection frequency, and experiencing homelessness. These associations and the limited uptake of harm reduction in this sample support continued efforts to expand harm reduction in this population. A key challenge in delivering services to this population is the high prevalence persons experiencing of homelessness which has not been previously demonstrated among PWID in India. PWID reporting homelessness, unstable housing, and migration may experience unmet needs for services and further disease transmission through socially and spatially dynamic networks.

A limitation of this cross-sectional analysis is that the reported networks and injection venues may not necessarily represent network members or venues where participants acquired HIV and/or HCV infection; however, this would likely bias observed associations toward the null and attenuated associations of sociometric and spatial factors. Further, the cross-sectional nature limits the ability to examine temporal associations, but the consistency of associations with active HCV infection suggests that these network factors may impact onward transmission. All responses related to drug use, network members, and spaces were self-reported and subject to social desirability and recall bias; to minimize bias, all interviewers were trained on optimal interviewing techniques. About 25% of referral coupons were not returned suggesting that the networks presented in these data may be incomplete; however, the response rate of 75% is higher than what has been seen in other network studies (*Kimani et al., 2014*; *Johnston et al., 2008*). PWID under the age of 18 were excluded from the study due to the legal age of consent in India; therefore, these individuals are not represented by the network topology. Statistical analyses assumed that observations are independent conditional on individual-level covariates. This assumption is likely to be violated to some extent, but violations are not expected to bias point estimates (they would result in underestimated standard errors).

Limitations notwithstanding, these data highlight the importance of networks on HIV and HCV burden in a community of PWID in New Delhi, India. Integrating strategies to intervene at sociometric- and spatial-levels in addition to individual-level interventions could improve the efficiency of prevention and treatment programming and may be critical to achieving epidemic control and elimination of HIV and HCV, respectively, while conserving resources.

## Acknowledgements

This study was supported by the National Institute on Drug Abuse of the National Institutes of Health (R01DA041736, DP2DA040244, R01DA041034, and K24DA035684), the Johns Hopkins University Center for AIDS Research (P30AI094189). The funders had no role in the design and conduct of the study; collection, management, analysis, and interpretation of the data; preparation, review, or approval of the manuscript; and decision to submit the manuscript for publication.

## Additional information

### Competing interests

Shruti H Mehta: reports personal fees from Gilead Sciences, outside the submitted work. Sunil S Solomon: reports grants/products and advisory board fees from Gilead Sciences and grants/products from Abbott Diagnostics, outside the submitted work. The other authors declare that no competing interests exist.

### Funding

| Funder | Grant reference number | Author |
| --- | --- | --- |
| National Institute on Drug Abuse | R01DA041736 | Sunil S Solomon |
| National Institute on Drug Abuse | DP2DA040244 | Sunil S Solomon |
| National Institute on Drug Abuse | R01DA041034 | Shruti H Mehta<br>Gregory M Lucas |
| National Institute on Drug Abuse | K24DA035684 | Gregory M Lucas |

The funders had no role in study design, data collection and interpretation, or the decision to submit the work for publication.

### Author contributions

Steven J Clipman, Conceptualization, Data curation, Software, Formal analysis, Validation, Investigation, Visualization, Methodology, Writing - original draft, Writing - review and editing; Shruti H Mehta, Conceptualization, Supervision, Funding acquisition, Methodology, Writing - original draft, Writing - review and editing; Aylur K Srikrishnan, Conceptualization, Supervision, Funding acquisition, Investigation, Project administration, Writing - review and editing; Katie JC Zook, Data curation, Software, Validation, Investigation, Project administration, Writing - review and editing; Priya Duggal, Supervision, Writing - review and editing; Shobha Mohapatra, Data curation, Supervision, Investigation, Project administration, Writing - review and editing; Saravanan Shanmugam, Paneerselvam Nandagopal, Investigation, Project administration, Writing - review and editing; Muniratnam S Kumar, Supervision, Funding acquisition, Investigation, Project administration, Writing - review and editing; Elizabeth Ogburn, Conceptualization, Formal analysis, Supervision, Validation, Methodology, Writing - review and editing; Gregory M Lucas, Conceptualization, Supervision, Funding acquisition, Methodology, Project administration, Writing - review and editing; Carl A Latkin, Conceptualization, Supervision, Methodology, Writing - review and editing; Sunil S Solomon, Conceptualization, Resources, Data curation, Supervision, Funding acquisition, Validation, Investigation, Methodology, Writing - original draft, Project administration, Writing - review and editing

### Author ORCIDs

Steven J Clipman https://orcid.org/0000-0002-2366-8420

### Ethics

Human subjects: The study protocol was approved by institutional review boards at Johns Hopkins Medicine (IRB00110421) and the YR Gaitonde Centre for AIDS Research and Education in India (YRG292). All participants provided written informed consent.

Decision letter and Author response
Decision letter https://doi.org/10.7554/eLife.69174.sa1
Author response https://doi.org/10.7554/eLife.69174.sa2

## Additional files

### Supplementary files

• Supplementary file 1. Baseline characteristics of 10 indexes that initiated network recruitment (parenthesis denote n unless otherwise specified).

• Reporting standard 1. STROBE Checklist for Cross-sectional Studies.

• Reporting standard 2. STROBE Flowchart.

• Transparent reporting form

### Data availability

An interactive version of the sociospatial network and underlying data are available from: https://github.com/sclipman/sociospatial-baseline.

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
