## [Decision Letter]

**Acceptance summary:**

This study explores the impact of social networks of injecting drug users on the risk of hepatitis C virus and HIV transmission. The study concludes that social and spatial networks of injecting drug users have a significant influence on their risk of becoming hepatitis C virus or HIV positive, on top of their individual risk factors. This paper is of interest for epidemiologists and those who are working for people who inject drugs and people living with HIV.

**Decision letter after peer review:**

Thank you for submitting your article "Role of direct and indirect social and spatial ties in the diffusion of HIV and HCV among people who inject drugs: A cross-sectional community-based network analysis in New Delhi, India" for consideration by *eLife*. Your article has been reviewed by 2 peer reviewers, and the evaluation has been overseen by a Reviewing Editor and Miles Davenport as the Senior Editor. The following individual involved in review of your submission has agreed to reveal their identity: Chaturaka Rodrigo (Reviewer #2).

Essential revisions:

1. In egocentric network, the connection between first degree alters is referred by an index, but such information may not accurate (alters may or may not connected regardless of the information from the index). Were these connections confirmed by the first degree alters? This is not just the issue in the information provided by indexes. It is generally challenging to confirm connections. Some may not share their names nor background because of various reasons beyond those described by the authors: 1. they do not need such information to share syringes, and 2. they are afraid that the information is transferred to law enforcement agencies, or 3. simply there are too many to refer (I do not think it is realistic to ask each person whether all 2511 participants are connected to him/her). The diameter and average path length were dramatically reduced by incorporating injection location, which might be explained by the incompleteness of the network, because undocumented connections could be explained by locations as the authors described.

2. The authors used "number infected injection partners" and "network distance from an HIV/HCV RNA^+^ Person" as predictor of HIV/HCV infection. It was not clear why these two factors which summarize the network structure were selected, although they sound reasonably associated with infection risk.

3. Is it possible to use different shapes to distinguish the participants and the hotspots?

4. Table 1 – Please explain the meaning of "anti-HCV" as a footnote in the table.

5. Table 1 – What is the relevance of "Ever used non-injecting drugs"? If this information needs to be put here, it is much better to state non-jecting drug use in the preceding six months.

6. "total 155 (86%) of the 181 injection locations included at least one HIV positive person" – This sentence is unclear.

7. "Individual-level variables associated with prevalent HIV in multivariable logistic regression included age, education, homelessness, sexual activity, needle sharing, injection frequency, and type of drug injected" – It is better to indicate the direction of association in this sentence (e.g. younger age is associated with HIV). While the details can be worked out from table 2, it is easier for the reader if this is mentioned here.

8. If a person had anti-HCV antibodies but no HCV RNA, did you ask about treatment?

9. "persons with active HCV infection had significantly higher degree centrality" – Is this for all edges (HCV and HIV) or edges from HCV infected nodes only?

10. Lines 175 – 181: Please provide adjusted OR with 95% CI for each % value as you have done for HIV section.

11. Line 241 – What is "Deep chain RDS"?

12. People under 18 has been excluded due to legal age of consent- This can have a significant effect on the network topology – please acknowledge this as a limitation.

13. Please provide sensitivity, specificity, NPV (negative predictive value) and PPV of tests you have used to diagnose HCV and HIV.

14. It is next to impossible to understand what you meant in the Results section by "Median network distance to another HIV antibody-positive PWID was 0", without the following explanation that appears in methods – "Network distance from an infected alter was calculated such that a distance of zero signifies a direct connection, a distance of one signifies one uninfected person along the shortest path between a given participant and infected alter" The latter information should also appear in results where you discuss median network distance.

15. Will there be any legal implications in publishing the map in Supplementary figure 3 – It may reveal the location of injecting hotspots to law enforcement, drive injection networks underground, and hinder future work of this cohort study. It will also have a negative impact on HCV and HIV prevention strategies on these locations.

16. Can another dimension be incorporated into the networks to indicate the impact of preventive measures over time? – This is a suggestion for future work and not necessarily to be addressed in this paper.*Reviewer #1 (Recommendations for the authors):*

The authors investigated the association between HIV/HCV infection risk and network structure (closeness to those infected). To rigorously investigate the network structure, they employed biometric data (fingerprint) along with identifying information of referred persons. They further coupled the network data with the information of where the participants inject drugs, which is to complement the network structure as well as to investigate the unique contribution of the special information to the infection risk. They found both network structure and spatial information explain infection risk even after controlling for individual-level risk factors. There are some studies investigated the network structure and its contribution to the infection risk of HIV/HCV, however, this study is unique as they used rigorous approach to reconstruct the network structure beyond egocentric network with substantial effort. This study gives insights to the network-based intervention to control HIV/HCV epidemics implemented around the world.

The strength of this study is that they recruited large number of participants and the effort to complete the network: they collected the information of network carefully avoiding duplicates using not just identifying information but also biometric data (fingerprint). Meanwhile, the limitation of this study is, as is described by the authors, incompleteness of the network.*Reviewer #2 (Recommendations for the authors):*

This study was well designed to achieve its objective of uncovering a broader social and spatial network of injecting drug users in New Delhi, India. The extent of the network uncovered is impressive starting with just 10 index cases. The reliability of data is high given that 75% of referrals distributed through this network was returned to investigators. The authors have maintained rigorous standards in methodology as far as practically possible for a study of this scale conducted in one of the most densely populated places on earth.

The methods explained here in my opinion, will set a new standard in research that explores a person's risk of blood borne viral infections (HCV/HBV and HIV). The results show that in addition to typically reported individual risk factors such as age and education etc, the social and spatial network of an injecting drug user also significantly influences their risk of HIV/HCV infection. This finding will be a boost to community-based prevention methods by providing evidence to back their existence and relevance.

The study has some limitations – findings are not generalisable outside the geography studied but these methods can be employed to study other areas of interest. The study has excluded persons under the age of 18, which can have a significant impact on the topology of the network in the form of missing links.

---

## [Author Response]

Essential revisions:1. In egocentric network, the connection between first degree alters is referred by an index, but such information may not accurate (alters may or may not connected regardless of the information from the index). Were these connections confirmed by the first degree alters? This is not just the issue in the information provided by indexes. It is generally challenging to confirm connections. Some may not share their names nor background because of various reasons beyond those described by the authors: 1. they do not need such information to share syringes, and 2. they are afraid that the information is transferred to law enforcement agencies, or 3. simply there are too many to refer (I do not think it is realistic to ask each person whether all 2511 participants are connected to him/her). The diameter and average path length were dramatically reduced by incorporating injection location, which might be explained by the incompleteness of the network, because undocumented connections could be explained by locations as the authors described.

We apologize for any confusion, however, the connections between first degree alters is not referred by an index as the reviewer suggests i.e., an index did not determine if two of their partners should be connected to each other. In this study network connections were only established by return of a referral card. Sampling was initiated with 10 indexes. The indexes were asked to name individuals who they injected with in the prior month (regardless of whether they shared injection paraphernalia) as well as provide a factoid about each individual (e.g., “he is originally from Bihar” or “she has a scar under the left eye”). They were then provided unique barcoded referral cards for each of the injection partners they named (there was no limit on the number of partners one could name), and they were asked to provide the card(s) to their partner(s). When an individual returned a referral card, a biometric fingerprint scan was used to determine if they had been previously enrolled in the study. If previously enrolled, the referral card was used to establish an additional cross-network link between the individual and person who referred them. If the individual was not previously enrolled, the factoid previously provided was used to confirm they were indeed the partner described and the referral card was used to establish a network connection with the participant that referred them. This recruited participant was next considered an “index” and in turn asked to name and provide a factoid about the persons they injected with in the prior month and was provided referral cards for each of these partners. They were then asked to distribute the cards to their partners and when the cards were returned, network connections were established and the process was repeated. Therefore, for a network connection (edge) to be created between two participants (nodes) there had to be a direct barcoded referral card linking the two participants; connections were not inferred by asking an index if his/her alters injected with each other. Moreover, cross-network linkages between individuals referred by different indexes were established via biometric linkage. Although 75% of all referral cards distributed were returned, we agree with the reviewer that additional undocumented connections may exist for a multitude of reasons. The key reasons raised by the reviewer include:

1) they do not need such information to share syringes and (2) they are afraid that the information is transferred to law enforcement agencies, or (3) simply there are too many to refer (I do not think it is realistic to ask each person whether all 2511 participants are connected to him/her).

Participants were asked to provide referral cards to individuals regardless of whether they share syringes and did not need to provide real names in the case they did not know the person well or were afraid that information would be transferred to law enforcement agencies. Our research group has spent over 10 years developing trust and rapport with this PWID community. We conducted thorough ethnographic research and met with community leaders and PWID across the city, and this was not a concern expressed by participants. Individuals may not recall all their injection partners; however, we asked several additional questions to get estimates of the number of PWID individuals know and have injected with and trained interviewers to clarify and revisit questions in the case of inconsistencies. Moreover, prior work across India has supported that the number of injection partners within the prior month is relatively small- a median of 2 partners. This is fairly consistent across cities where we have worked in India. Moreover, we were asking participants to recall people they specifically injected with not just had passing contact with. We used the past month to minimize these recall issues.

Additionally, 75% network completeness is high for network studies among PWID, and we believe that the addition of spaces individuals inject allow us to further model undocumented connections. The network diameter and average path length were reduced by incorporating injection location, demonstrating the utility of spaces in reaching network participants, and we agree with the reviewer that this might be explained by the incompleteness of the network, but it also does not strictly signify incompleteness. Given a toy example where all injection partner connections are known, creating a bi-partite network with spaces can still result in reduced pathlength and diameter, irrespective of completeness. For example, in Author response image, 1 of a hypothetical example of a complete network, the diameter of (a) is 3 since the longest shortest paths have length of 3 edges (e.g., between 6-1 and 6-2); however, if we create a bi-partite network with a common space depicted as a blue square in (b), the diameter is reduced to 2.

**Author response image 1. sa1fig1:** 

2. The authors used "number infected injection partners" and "network distance from an HIV/HCV RNA^+^ Person" as predictor of HIV/HCV infection. It was not clear why these two factors which summarize the network structure were selected, although they sound reasonably associated with infection risk.

We considered candidate risk factors based on biological/epidemiological significance and prior literature and also used a Boruta random forest feature selection algorithm to comprehensively explore candidate factors. Both the number of infected partners and network distance to an infected partner had significant variable importance scores from random forest (p<0.05) and are epidemiologically meaningful since an increase in these variables represents an increased number of pathways through which transmission could occur, as demonstrated in several treatment as prevention studies for HIV. While existing studies have reported associations with HIV/HCV infection and the number of infected partners, network distance to an infected partner may help to explain the increased risk of transmission even before a direct connection (injection partner) is diagnosed or represent viremia in the extended partner network, further approximating the chance that an unreported or future injection partner is infected, which to our knowledge is novel.

3. Is it possible to use different shapes to distinguish the participants and the hotspots?

We thank the reviewer for this excellent suggestion. The figure has been revised to display injection locations as squares.

4. Table 1 – Please explain the meaning of "anti-HCV" as a footnote in the table.

We have added the meaning of anti-HCV as a footnote in the table.

5. Table 1 – What is the relevance of "Ever used non-injecting drugs"? If this information needs to be put here, it is much better to state non-jecting drug use in the preceding six months.

Thank you, we have removed "Ever used non-injecting drugs" from table 1.

6. "total 155 (86%) of the 181 injection locations included at least one HIV positive person" – This sentence is unclear.

We have revised this sentence in the manuscript to read:

“At least one HIV-positive person reported injecting at 155 (86%) of the 181 injection locations identified by participants.”7. "Individual-level variables associated with prevalent HIV in multivariable logistic regression included age, education, homelessness, sexual activity, needle sharing, injection frequency, and type of drug injected" – It is better to indicate the direction of association in this sentence (e.g. younger age is associated with HIV). While the details can be worked out from table 2, it is easier for the reader if this is mentioned here.

Thank you, we appreciate this suggestion and have revised the manuscript to indicate the direction of association in this sentence. The revised sentence reads:

“Individual-level variables positively associated with prevalent HIV in multivariable logistic regression included younger age, lower education, experiencing homelessness, decreased sexual activity, sharing syringes, increased injection frequency, and injecting heroin and buprenorphine (Table 2).”

8. If a person had anti-HCV antibodies but no HCV RNA, did you ask about treatment?

We asked all participants about prior HCV testing and treatment. The majority of these PWID were unaware of their HCV status, only 104 (4%) had ever been previously tested for HCV; 9 individuals reported ever testing positive (all of whom received treatment or were currently taking treatment), and 3 reported a sustained virologic response. Therefore, instances where a person had anti-HCV antibodies but no HCV RNA most likely represent natural/spontaneous clearance of HCV infection. We have revised the manuscript to include this information.

9. "persons with active HCV infection had significantly higher degree centrality" – Is this for all edges (HCV and HIV) or edges from HCV infected nodes only?

We used the standard definition of degree centrality i.e., the total number of edges a node has, irrespective of HIV and/or HCV status.

10. Lines 175 – 181: Please provide adjusted OR with 95% CI for each % value as you have done for HIV section.

We thank the reviewer for catching this and have revised the manuscript to include the adjusted OR with 95% CI for each value.

11. Line 241 – What is "Deep chain RDS"?

We thank the reviewer for pointing out that the acronym for respondent-driven sampling (RDS) was not previously defined. We have revised the manuscript to define RDS and have also rearranged the citations in the sentence to better detail which reference corresponds to “deep chain RDS” should the reader be interested in learning more. “Deep chain RDS” is an approach that hypothesizes that the deeper the recruitment depth in a RDS sample, the more likely one is to identify HIV+ persons not engaged in care.

12. People under 18 has been excluded due to legal age of consent- This can have a significant effect on the network topology – please acknowledge this as a limitation.

We appreciate the reviewer highlighting this and have revised the manuscript to acknowledge this as a limitation.

13. Please provide sensitivity, specificity, NPV (negative predictive value) and PPV of tests you have used to diagnose HCV and HIV.

We have added details on the sensitivity and specificity for each test used to diagnose HCV and HIV to the Materials and methods section.

14. It is next to impossible to understand what you meant in the Results section by "Median network distance to another HIV antibody-positive PWID was 0", without the following explanation that appears in methods – "Network distance from an infected alter was calculated such that a distance of zero signifies a direct connection, a distance of one signifies one uninfected person along the shortest path between a given participant and infected alter" The latter information should also appear in results where you discuss median network distance.

We thank the reviewer for pointing this out. The article structure has been reordered so that methods now come before the results, which should remedy this.

15. Will there be any legal implications in publishing the map in Supplementary figure 3 – It may reveal the location of injecting hotspots to law enforcement, drive injection networks underground, and hinder future work of this cohort study. It will also have a negative impact on HCV and HIV prevention strategies on these locations.

We understand and acknowledge the reviewer’s concern and have further anonymized this figure (Figure 3—figure supplement 1). We have replaced the map with a solid white background so injection locations cannot be geographically placed. The distance between points is still geospatially congruent to give the reader an idea of their distribution and HIV/HCV prevalence.

16. Can another dimension be incorporated into the networks to indicate the impact of preventive measures over time? – This is a suggestion for future work and not necessarily to be addressed in this paper

Thank you for this excellent suggestion, we are currently in the process of modeling the impact of various preventive measures over time and plan to publish an additional paper on these findings.